# Long-Term Outcomes of Congenital Diaphragmatic Hernia: Report of a Multicenter Study in Japan

**DOI:** 10.3390/children9060856

**Published:** 2022-06-08

**Authors:** Masaya Yamoto, Kouji Nagata, Keita Terui, Masahiro Hayakawa, Hiroomi Okuyama, Shoichiro Amari, Akiko Yokoi, Kouji Masumoto, Tadaharu Okazaki, Noboru Inamura, Katsuaki Toyoshima, Yuhki Koike, Yuta Yazaki, Taizo Furukawa, Noriaki Usui

**Affiliations:** 1Department of Pediatric Surgery, Shizuoka Children’s Hospital, Shizuoka 420-8660, Japan; ped.surg1018@gmail.com; 2Department of Pediatric Surgery, Reproductive and Developmental Medicine, Faculty of Medical Sciences, Kyushu University, 3-1-1, Maidashi, Higashi-Ku, Fukuoka 812-8582, Japan; 3Department of Pediatric Surgery, Chiba University Graduate School of Medicine, Chiba 260-867, Japan; kta@cc.rim.or.jp; 4Division of Neonatology, Center for Maternal-Neonatal Care, Nagoya University Hospital, Nagoya 466-8560, Japan; masahaya@med.nagoya-u.ac.jp; 5Department of Pediatric Surgery, Osaka University Graduate School of Medicine, Osaka 565-0871, Japan; okuyama@pedsurg.med.osaka-u.ac.jp; 6Division of Neonatology, National Center for Child Health and Development, Tokyo 157-0074, Japan; amari-s@ncchd.go.jp; 7Department of Pediatric Surgery, Kobe Children’s Hospital, Kobe 650-0047, Japan; yokoi_kch@hp.pref.hyogo.jp; 8Department of Pediatric Surgery, Tsukuba University, Tsukuba 305-8577, Japan; kmasu@md.tsukuba.ac.jp; 9Department of Pediatric Surgery, Juntendo University Urayasu Hospital, Urayasu 279-0021, Japan; okazakit@juntendo.ac.jp; 10Department of Pediatrics, Kindai University Faculty of Medicine, Osaka-Sayama 589-0014, Japan; ki88mi@gmail.com; 11Department of Neonatology, Kanagawa Children’s Medical Center, Yokohama 232-8555, Japan; nqf37179@nifty.com; 12Department of Gastrointestinal and Pediatric Surgery, Mie University Graduate School of Medicine, Tsu 514-8507, Japan; koikyon@gmail.com; 13Department of Pediatric General and Urogenital Surgery, Juntendo University School of Medicine, Tokyo 113-8421, Japan; yyazaki@juntendo.ac.jp; 14Department of Pediatric Surgery, Graduate School of Medical Science, Kyoto Prefectural University of Medicine, Kyoto 602-8566, Japan; taizow@koto.kpu-m.ac.jp; 15Department of Pediatric Surgery, Osaka Women’s and Children’s Hospital, Izumi 594-1101, Japan; usui@wch.opho.jp

**Keywords:** congenital diaphragmatic hernia, long-term outcomes, recurrence, pneumonia, pneumothorax, gastroesophageal reflux disease, intestinal obstruction

## Abstract

Background: Treatment modalities for neonates with congenital diaphragmatic hernia (CDH) have greatly improved in recent years, with a concomitant increase in survival. However, long-term outcomes restrict the identification of optimal care pathways for CDH survivors in adolescence and adulthood. Therefore, we evaluated the long-term outcomes within the Japanese CDH Study Group (JCDHSG). Methods: Participants were born with CDH between 2006 and 2018 according to the JCDHSG. Participants were enrolled in the database at 1.5, 3, 6, and 12 years old. Follow-up items included long-term complications, operations for long-term complication, and home medical care. Results: A total of 747 patients were included in this study, with 626 survivors (83.8%) and 121 non-survivors (16.2%). At 1.5, 3, 6, and 12 years old, 45.4%, 36.5%, 34.8%, and 43.6% developed complications, and 20.1%, 14.7%, 11.5%, and 5.1% of participants required home care, respectively. Recurrence, pneumonia, pneumothorax, gastroesophageal reflux disease, and intestinal obstruction decreased with age, and thoracic deformity increased with age. Conclusions: As CDH survival rates improve, there is a need for continued research and fine-tuning of long-term care to optimize appropriate surveillance and long-term follow-up.

## 1. Introduction

Congenital diaphragmatic hernia (CDH) is a developmental abnormality, with the diaphragmatic defect occurring in conjunction with pulmonary, cardiac, gastrointestinal, neurodevelopmental, and musculoskeletal impairments [1,2]. Overall survival rates for CDH have improved in recent decades with intensive research and the use of standardized postnatal management protocols, including gentle ventilation strategies, extracorporeal membrane oxygenation therapy (ECMO), ongoing pulmonary hypertension treatment, and referral of patients to high-volume centers [3].

However, modern survivors of CDH bear the potential for increased morbidity, as CDH patients are at risk for long-term morbidity, including chronic pulmonary dysfunction, persistent reactive pulmonary vascular bed, neurodevelopmental disorders, hearing loss, nutritional disorders, musculoskeletal abnormalities, and more. Patients with CDH require long-term medication therapy, home respiratory support, and often multiple subsequent surgical interventions. These needs continue to affect survivors from infant to adulthood.

Therefore, we explored the key components of long-term follow-up for CDH survivors in Japan.

## 2. Materials and Methods

### 2.1. Patient Selection

The Japanese CDH Study Group (JCDHSG) conducted a multicenter cohort study that had a retrospective design between 2006 and 2016 and has been expanded with a prospective cohort study since 2017. This database includes infants with CDH who were admitted to 15 participating centers between January 2006 and December 2018 and diagnosed prenatally or within 28 days after birth. In this study, we reviewed the JCDHSG database. Participants were enrolled in the database at 1.5, 3, 6, and 12 years old.

### 2.2. Patient Variables

A retrospective chart review was performed to collect demographic information as well as data on initial admission, including patient characteristics (gender, birth weight, gestational age, comorbidities including genetic, cardiac and neurological disorder), details concerning the diagnosis, disease severity, history of fetal treatment, type of delivery, operative approach, repair method, history of nitric oxide (NO), and history of ECMO.

### 2.3. Outcomes

Follow-up items included long-term complications (recurrence, persistent pulmonary hypertension (PPH), pneumonia, pneumothorax, gastroesophageal reflux disease (GERD), intestinal obstruction, thoracic deformity, hearing loss, undescended testis), operations for long-term complication (repair for recurrence, fundoplication, surgery for intestinal obstruction, surgical correction for thoracic deformity, orchiopexy), and home medical care (oxygen, ventilator, tracheostomy, enteral nutrition, parenteral nutrition).

### 2.4. Statistical Analysis

A descriptive analysis of the characteristics and long-term outcomes was performed using the JMP software program (version 12.01; SAS Institute, Inc., Cary, NC, USA). Continuous variables are presented as medians with interquartile ranges (IQRs), while categorical variables are presented as totals and percentages.

## 3. Results

### 3.1. Clinical Characteristics and Mortality

Overall, 747 patients were included in the JCDHSG database during the study period, of whom 626 (83.8%) survived, while 121 (16.2%) died by 1.5 years old. There were 626, 519, 264, and 73 participants enrolled at 1.5, 3, 6, and 12 years old, respectively.

Table 1 shows the demographic data. Of these patients, 340 (54.3%) were male and 286 (45.7%) were female. The median gestational age was 37 (IQR 37–38) weeks and the median birth weight was 2724 (IQR 2426–2990) g, with a genetic variant diagnosed in 2.7%, cardiac malformation in 9.4%, and neurological disorder in 1.8% of patients. The most commonly affected side for CDH was the left side (92.5%), followed by the right side (7.3%) and bilateral hernias (0.2%). The diaphragmatic defect was >50% of the chest wall in 30.8% of patients, and liver elevation was present in 33.1% of patients. Here, 6 patients underwent fetal endoscopic tracheal occlusion as fetal therapy. Furthermore, 64.2% of patients underwent a cesarean section. The median age at repair was 2 (IQR 2–3) days. The most common surgical approach was open with an abdominal approach (88.4%) followed by a thoracoscopic approach (10.1%). Patients underwent repair, with 34.7% being repaired with a patch and 64.2% undergoing primary repair. ECMO and NO support were required in 4.6% and 67.1% of patients, respectively.

### 3.2. Recurrence of CDH

The patients were observed up to 1.5, 3, 6, and 12 years old, with 58 of 626 (9.3%), 11 of 516 (2.1%), 5 of 264 (1.9%), and 0 of 73 (0%) patients having CDH recurrence, respectively (Figure 1). Fifty-seven of 626 patients (9.1%) required surgery for recurrence up to 1.5 years, 5 of 516 patients (1.0%) at 1.5 to 3 years, 4 of 264 patients (1.5%) at 3 to 6 years, and 0 of 73 (0%) patients at 6 to 12 years, respectively (Figure 2). Fifty-nine of 516 patients (11.4%) had recurrent CDH by the age of 3 years, requiring surgical repair, including 41 (69.4%) who had a history of patch repair.

### 3.3. Long-Term Cardiopulmonary Complications

Thirty-six of 626 patients (5.8%) required medication at 1.5 years, 27 of 516 patients (5.2%) at 3 years, 3 of 264 patients (1.9%) at 6 years, and 1 of 73 patients (1.4%) at 12 years, respectively (Figure 1). Fifty-five of 626 patients (8.8%) required admission for pneumonia up to 1.5 years, 37 of 516 patients (7.2%) at 1.5 to 3 years, 14 of 264 patients (5.3%) at 3 to 6 years, and 2 of 73 (2.7%) patients at 6 to 12 years (Figure 1). Furthermore, 56 of the 626 patients developed pneumothorax before the age of 1.5 years. No patient developed pneumothorax between the ages of 1.5 and 12 years (Figure 1). At 1.5, 3, 6, and 12 years old, 8.3%, 5.0%, 3.8%, and 2.7% of participants needed oxygen; 2.9%, 1.7%, 1.9%, and 0% of participants required a ventilator; and 3.5%, 2.9%, 2.7%, and 2.7% of participants required tracheostomy, respectively (Figure 3).

### 3.4. Long-Term Gastrointestinal Complications

A total of 109 of 626 patients (17.4%) required medication for GERD up to 1.5 years, 46 of 516 patients (8.9%) at 1.5 to 3 years, 18 of 264 patients (6.8%) at 3 to 6 years, and 2 of 73 (2.7%) patients at 6 to 12 years (Figure 1). Here, 48 of 626 patients (7.7%) required fundoplication for GERD up to 1.5 years, 14 of 516 patients (2.7%) at 1.5 years to 3 years, 2 of 264 patients (0.8%) at 3 to 6 years, and 1 of 73 (1.4%) patients at 6 to 12 years (Figure 2).

Here, 65 (10.4%) and 41 (6.5%) of 626 patients required treatment and operation for intestinal obstruction up to 1.5 years, 23 (4.5%) and 15 (2.9%) of 516 patients at 1.5 years to 3 years, 13 (4.9%) and 6 (2.3%) of 264 patients at 3 to 6 years, and 2 (2.7%) and 1 (1.4%) of 73 patients at 6 to 12 years, respectively (Figure 1 and Figure 2).

At 1.5, 3, 6, and 12 years old, 8.0%, 5.0%, 3.8%, and 2.7% of participants needed enteral nutrition, and 2.2%, 2.9%, 3.0%, and 1.4% of participants required parenteral nutrition, respectively (Figure 3).

### 3.5. Long-Term Other Complications

Here, 101 (16.1%) and 1 (0.2%) of 626 patients were diagnosed and required operation for thoracic deformity up to 1.5 years, 94 (18.2%) and 1 (0.2%) of 516 patients at 1.5 years to 3 years, 50 (19.0%) and 1 (0.4%) of 264 patients at 3 to 6 years, and 20 (27.4%) and 2 (2.7%) of 73 patients at 6 to 12 years, respectively (Figure 1 and Figure 2).

Twenty-eight of 626 patients (4.5%) diagnosed of hearing loss up to 1.5 years, 18 of 516 patients (3.5%) at 1.5 years to 3 years, 9 of 264 patients (3.4%) at 3 to 6 years, and 1 of 73 (1.4%) patients at 6 to 12 years (Figure 1).

Twenty-nine (9.1%) and 19 (5.9%) of 320 male patients were diagnosed and required orchiopexy for undescended testis up to 1.5 years, 13 (4.8%) and 11 (4.1%) of 270 male patients at 1.5 years to 3 years, 3 (2.0%) and 2 (1.3%) of 150 male patients at 3 to 6 years, and 1 (2.8%) and 1 (2.6%) of 38 male patients at 6 to 12 years, respectively (Figure 1 and Figure 2).

## 4. Discussion

CDH is a rare congenital anomaly caused by a defect in the pleuroperitoneal membrane. The survival rate of patients with CDH has been increasing over the past three decades [3]. As survival lengthens, it becomes increasingly important to focus on the long-term comorbidities associated with CDH. Many studies have documented the diversity and frequency of long-term sequelae involving the cardiopulmonary, gastrointestinal, neurodevelopmental, and musculoskeletal systems [4,5,6]. We herein provide an overview of the long-term outcomes of JCDHSG and offer suggestions for future studies.

### 4.1. Recurrence

Vomiting, intestinal obstruction, abdominal pain, and respiratory distress are the most common complaints of hernia recurrence. In the studies that mentioned recurrence, the recurrence rates ranged from 2 to 25% [7,8,9,10,11]. In the present study, recurrence occurred in 4.3% of cases by the time of discharge, 7.2% between discharge and 1.5 years old, 2.1% between 1.5 and 3 years old, and 1.9% after 3 years old.

The recurrence rate reportedly varies depending on the severity of the disease, the size of the defect, and the presence or absence of a patch, with an increased defect size being associated with an increased risk of recurrence. The recurrence rate of direct suturing was reported to be about 4%, as compared to the recurrence rate of patch repair, which was 27–41% [12,13]. In our study, patch repair appeared to be associated with recurrence, as almost 70% of patients who experienced recurrence initially required a patch. This may be due to the severity of CDH in these patients, as the patch is primarily used in patients with large defects.

In a systematic review of retrospective studies concerning thoracoscopic surgery for CDH, the recurrence rate has been reported to be high compared to open surgery [14]. However, this is controversial because of the high recurrence rate, selection bias, and lack of stratification when comparing results. Recently, a propensity-score-matched study on recurrence rates found no significant difference in these rates between open and thoracoscopic surgery [15].

### 4.2. Cardiopulmonary Complications

#### 4.2.1. PPH

PPH contributes to the mortality of CDH patients and requires home oxygen therapy, tracheostomy, and a ventilator for pulmonary hypertension. The incidence rates of postoperative PPH ranged from 3 to 12% in long-term observations [16,17,18]. Pulmonary hypertension typically resolves within weeks to months after surgical repair in most CDH patients. In fact, the need for a vasodilator to manage PPH decreased after six years in this study; however, there have been reports of its persistence or reoccurrence in adolescence, with some cases being clinically asymptomatic [11,19,20]. Evidence concerning recommendations for long-term follow-up was scarce in the literature and mostly suggested by expert opinion [4,11]. However, the American Academy of Pediatrics provided a template for follow-up of CDH patients after discharge, recommending echocardiography and cardiology follow-up every 3 months for the first 18 months, followed by supplemental oxygen or an annual evaluation if the previous echocardiogram was abnormal in 2008 [21]. Subsequently, the American Heart Association and the American Thoracic Society proposed the Class I Level of Evidence B guidelines for the routine evaluation of patients with CDH in 2015, which included long-term care in an interdisciplinary PH program and subsequent recommendations provided to all children with PPH [22]. Potential PPH may present into childhood, so a balanced follow-up plan should be considered to avoid missed diagnoses [20].

#### 4.2.2. Pneumonia

The incidence rates of postoperative respiratory complications, including pneumonia, pneumothorax and asthma, ranged from 10% to 50% according to long-term observation [23,24,25]. In particular, chronic recurrent pneumonia may be an additional morbidity that occurs in CDH survivors; a subset of CDH survivors may develop pneumonia up to several times per year, and these patients are likely to have a large defect requiring high-frequency oscillatory ventilation (HFOV) and long-term artificial ventilation [26]. In the present study, 8.8% of survivors required hospitalization for pneumonia by 1.5 years. The incidence of pneumonia in our CDH patients was less than previously reported in the literature, due to the admission cases for pneumonia only being enrolled in the database. Pneumonia in infancy needs to be carefully managed. For immunization in infants with CDH, the American Academy of Pediatrics recommends the administration of palivizumab (respiratory syncytial virus monoclonal antibody, Synagis©; AstraZeneca, Wilmington, DE, USA) [1].

### 4.3. Gastrointestinal Complications

#### 4.3.1. GERD

Most CDH survivors develop GERD due to the deviation and shortening of the abdominal esophagus, closure under excessive tension of the diaphragmatic defect, esophageal kinking at the gastroesophageal junction, increased pressure gradient across the hiatus, and the absence of a perihilar diaphragm [27,28]. Various predictors of GERD have been reported, including herniation of the liver or stomach, size of the defect, patch repair, and use of ECMO [29,30]. The risk of GERD in CDH is reported to have increased with the improvement of the survival rate in cases of severe CDH [31,32]; nearly 50% of CDH survivors are thought to have symptomatic GERD [33]. However, the natural history of GERD associated with CDH reportedly tends to resolve without surgery, and similar self-resolution is seen in isolated GERD [34,35]. In the present study, GERD occurred in 17.4% of patients by age 1.5 years and in 7.7% of all cases that underwent surgery, although 60% of patients who received medical therapy for GERD were completely cured by the age of 3 years. Among CDH survivors, GERD cases that require anti-reflux surgery and cases with potential self-resolution without surgery were mixed. Due to the high prevalence of GERD, the severity of its sequelae, and the fact that many patients with CDH require anti-reflux procedures, some authors have proposed the use of prophylactic wraps at the time of CDH repair [30,36]. However, one randomized controlled trial on this issue found no long-term benefit from performing wraps at the time of CDH repair [37].

#### 4.3.2. Intestinal Obstruction

It has been previously reported that intestinal obstructions are the major cause of adhesions in infants [12,38]. In the present study, 10.4% of all CDH patients required treatment for intestinal obstruction up to 1.5 years, 4.5% at 1.5 years to 3 years, 4.9% at 3 to 6 years, and 2.7% at 6 to 12 years. Adhesive intestinal obstructions were significantly more common in patients with patch repair than repair without a patch, which may indicate that the patch used influenced the formation of adhesions. In addition, after bowel repositioning, the formation of adhesions may occur, and the severity of adhesions may be linked to the amount of bowel handling during surgery. While the association of intestinal malrotation and CDH has been previously described, with incidence percentages ranging from 42 to 60%, the incidence rate of midgut volvulus was 1.5% [39,40]. In addition, a previous study showed that a preemptive Ladd procedure performed prophylactically in patients with CDH did not reduce subsequent volvulus [41]. Further studies are still needed to discuss the pros and cons of a preemptive Ladd procedure.

### 4.4. Other Complications

#### 4.4.1. Thoracic Deformity

Thoracic deformities, including scoliosis and pectus excavatum, may be due to asymmetrical lung development and have been associated with restrictive lung disease and increased tension at the diaphragm during operative repair [42,43]. The common musculoskeletal abnormalities were reported to be the pectus excavatum in 20% of CDH survivors and scoliosis in 30% of CDH survivors [44]. Thoracic deformities (36%) were diagnosed most often after patch repair, especially when ECMO was needed (60%) [45]. In our study, 16.1% of CDH survivors were diagnosed with a thoracic deformity up to 1.5 years, 18.2% at 3 years, 19.0% at 6 years, and 27.4% at 12 years. By the age of 3 years, about 70% of the patients had pectus excavatum and 20% had scoliosis, but by the age of 12 years about 40% had pectus excavatum and 60% had scoliosis, with the scoliosis becoming more evident as the years progressed. Surgery was performed in less than 10% of the cases diagnosed up to the age of 12. Thoracic deformities progress throughout the entire growth period and require long-term follow-up into adulthood [46].

#### 4.4.2. Hearing Loss

Some neonatal respiratory treatments are suggested to be risk factors for hearing loss in general. Hearing loss has also been identified in survivors of CDH. Furthermore, the rates of hearing impairment in patients with CDH range from 2% to 56% [6,47,48]. However, our study showed that 4.5% of patients were diagnosed with hearing loss by 1.5 years, 3.5% at 1.5 years to 3 years, 3.4% at 3 to 6 years, and 1.4% patients at 6 to 12 years. The incidence of hearing loss in our CDH patients was low compared to previous reports in the literature, did not tend to have a late onset, and was progressive. A previous report noted that hearing loss is associated with the need for ECMO and a prolonged duration of aminoglycoside treatment [49]. Children may be at an even greater risk for developing hearing loss if they have a history of congenital diaphragmatic hernia, prolonged ECMO therapy, or a lengthy course of aminoglycoside antibiotic therapy. As a result, they should be monitored closely throughout childhood, and their risk should be individually assessed.

#### 4.4.3. Undescended Testis

Undescended testis is a common congenital anomaly, with an estimated incidence range of 1.5% to 3% [50,51]. Some reports have reported a 25–30% incidence range of undescended testes in male patients with high-risk CDH [52,53,54]. In the present study, the incidence of undescended testes requiring orchiectomy in CDH was approximately 15%, which was about 6 times higher than in the general population [50,51]. There was also evidence of a correlation between undescended testis and CDH.

### 4.5. Limitations

The current study had several limitations. First, due to its multicenter design, the therapeutic strategies were not made uniform until 2017. Second, this study was a retrospective review, so patients may not have had complete data, necessitating all analyses to be based on patients with all required data available. Third, as noted in this and other studies, the incidence of the long-term prognosis in CDH patients varies with age. The follow-up periods for our patients ranged from 3 to 15 years, which may have influenced the reported results. Finally, patients with genetic variants, congenital cardiac malformations, or congenital neurological disorders are generally associated with long-term sequelae related to their morbidity; however, these patients were included in the analysis because of the small proportion of each sequelae.

## 5. Conclusions

In conclusion, we built upon our group’s long-term experience in CDH. As a result, we recognized that long-term follow up for CDH requires a multidisciplinary approach with consultations that include neonatology, pediatric pulmonology, surgery, cardiology, gastroenterology, neuropsychology, and nutrition. In the future, we plan to use the results of this study to determine the optimal surveillance strategy to enhance the efficiency and scope of multidisciplinary care.

## Figures and Tables

**Figure 1 children-09-00856-f001:**
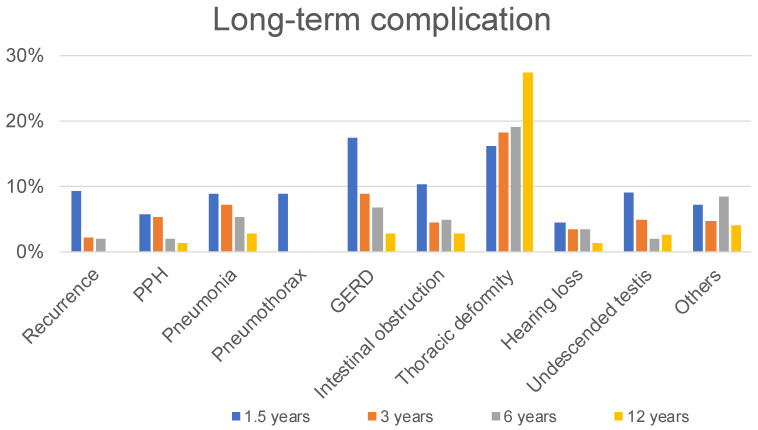
The proportions of long-term complications.

**Figure 2 children-09-00856-f002:**
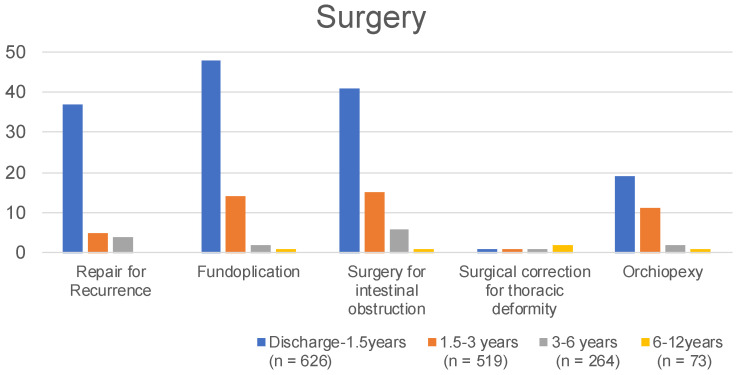
The proportions of different surgical procedures.

**Figure 3 children-09-00856-f003:**
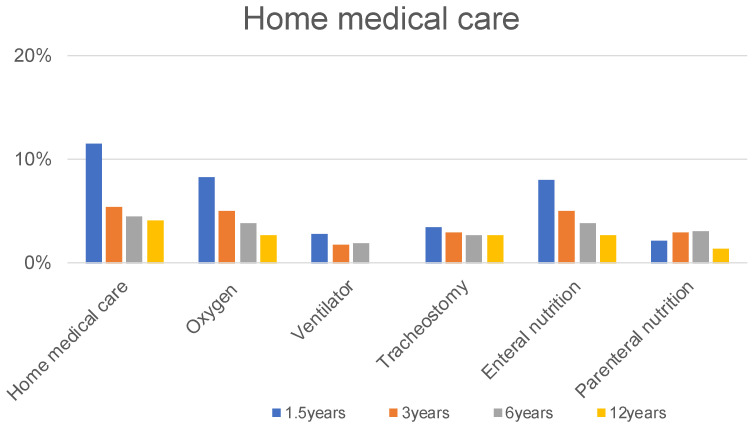
The proportions of home medical care and type.

**Table 1 children-09-00856-t001:** Demographic data and characteristics (N = 626).

Variables ^1^	N or Median	(% or IQR)
Gender		
Male	340	(54.3)
Female	286	(45.7)
Gestational age (week)	37	(37–38)
Birth weight (g)	2724	(2426–2990)
Genetic variant ^2^		
Mild	9	(1.4)
Severe	8	(1.3)
Cardiac malformation ^2^		
Mild	35	(5.6)
Severe	24	(3.8)
Neurological disorder ^2^		
Mild	5	(0.8)
Severe	3	(0.5)
CDH laterality		
Left	579	(92.5)
Right	46	(7.3)
Bilateral	1	(0.2)
Diaphragmatic defect size ^3^		
Defect size A and B	401	(64.1)
Defect size C and D	101	(30.8)
Unknown	32	(5.1)
Position of the liver		
Intra-abdominal	419	(66.9)
Intrathoracic	207	(33.1)
Fetal Endoscopic Tracheal Occlusion	6	(1)
Delivery		
Vaginal birth	224	(35.8)
Caesarean section	402	(64.2)
Age at repair (days)	2	(2–3)
Surgical approach		
Abdominal	554	(88.4)
Thoracic	6	(1)
Laparoscopy	3	(0.5)
Thoracoscopy	63	(10.1)
Repair method		
Primary closure	401	(64.2)
Patch closure	217	(34.7)
Muscular tissue	2	(0.3)
Unknown	5	(0.8)
History of ECMO ^4^	29	(4.6)
History of NO ^5^	420	(67.1)

^1^ Reported as the median (IQR; interquartile range) or n (%). ^2^ Severe was defined as affecting life prognosis, systemic status, and respiratory and circulatory dynamics, while mild was defined as strongly non-affecting life prognosis. ^3^ Based on the CDH Study Group Staging System, diaphragm defects were classified as defect size A, B, C, or D: Defect A, diaphragm defect involves <10% of the circumference of the chest wall; defect B, diaphragm defect involves <50% of the chest wall; defect C, diaphragm defect involves >50% of the chest wall; defect D, diaphragm defect involves >90% of the chest wall. ^4^ ECMO = extracorporeal membrane oxygenation. ^5^ NO = nitric oxide.

## Data Availability

The data presented in this study are available on request from the corresponding author. The datasets are stored at the University of Osaka and can be accessed there. The data are not publicly available due to the fact that this is not in accordance with consent provided by participants on the use of confidential data.

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
