# Peer review of "Long-Term Outcomes of Congenital Diaphragmatic Hernia: Report of a Multicenter Study in Japan"

_children, 2022, doi:10.3390/children9060856_

Round 1
Reviewer 1 Report
The paper analyses the late consequences of CDH repair in a large cohort of patients.
I would have liked to read something on the transition of care of these long surviving patients. The Authors should explain if such a service is present in Japan and how it is structured.
The analyzed population refers to a wide period of rime (11 years). Did the Authors analyze if differences in survival, recurrences, and other complications change during this long period . If not, they should mention it. If yes, they should break down the period into 2 (?) shorter periods and report the differences.
English needs some further reviewing. Here are just some examples: "...a balanced follow-up plan should be considered that avoids missing the diagnosis", "an patch", "...needs to home..", "...morbidity incurred in CDH survivors...", .."up several times per year per year..", "..was low cases.."
These sentences are unclear: "One of the causes could be the closure of the abdomen in patients with large defects who needed patch repair, which gives an increased intra-abdominal volume but may still give a high intra-abdominal pressure after surgery", "However, it remains controversial because of the high recurrence rate, selection bias, and lack of stratification when comparing results", "We reported the study which performed propensity score matching was no significant difference between..", "As is important with all morbidities in this patient population..."
Author Response
Author's Reply to the Review Report (Reviewer 1)
Thank you for the opportunity of revising our manuscript and considering it for publication in Children. We would like to thank the reviewers for their constructive comments.
Please find attached our revised manuscript and response point-by-point to the reviewers.
We feel that this article will be appealing to the readership of the journal as it is relevant to CDH and Pediatrics in general.
We confirm that the manuscript has not been published previously.
We look forward to your comments and suggestions and thank you in advance for your consideration.
The paper analyses the late consequences of CDH repair in a large cohort of patients.
I would have liked to read something on the transition of care of these long surviving patients. The Authors should explain if such a service is present in Japan and how it is structured.
The analyzed population refers to a wide period of rime (11 years). Did the Authors analyze if differences in survival, recurrences, and other complications change during this long period . If not, they should mention it. If yes, they should break down the period into 2 (?) shorter periods and report the differences.
→In Japan, each institution has conducted treatment and long-term follow-up based on its experience. As a result, the survival rate has remained the same over the past 10 years. Therefore, there is little significance in examining the two historical groups separately, and the paper is positioned to present the treatment results and long-term prognosis in Japan at the present time. In 2018, the JCDHSG developed a treatment protocol, including long-term follow-up, and we would like to evaluate future results in the future.
English needs some further reviewing. Here are just some examples: "...a balanced follow-up plan should be considered that avoids missing the diagnosis", "an patch", "...needs to home..", "...morbidity incurred in CDH survivors...", .."up several times per year per year..", "..was low cases.."
These sentences are unclear: "One of the causes could be the closure of the abdomen in patients with large defects who needed patch repair, which gives an increased intra-abdominal volume but may still give a high intra-abdominal pressure after surgery", "However, it remains controversial because of the high recurrence rate, selection bias, and lack of stratification when comparing results", "We reported the study which performed propensity score matching was no significant difference between..", "As is important with all morbidities in this patient population..."
→Sorry for various English mistakes. We have rewritten and proofread the English text and would appreciate your review.
Sincerely yours,
Masaya Yamoto
Kouji Nagata

Reviewer 2 Report
Thank you very much for the opportunity to review this manuscript that I read with great interest. In this retrospective review of database from JCDHSC, investigators reported long-term outcome of infants diagnosed with CDH either antenatally or during the first 4 weeks after birth. Investigators reported incidence of different complications associated with CDH infants at 1.5, 3, 6, and 12 years of age.
I would like to congratulate the investigators for the well written manuscript
Here are my comments on the manuscript
1) Title: reflect the study content
2) Abstract: summarizes well the study methodology and results
3) Introduction: concise and indicated aim of the study
4) Methods: No concerns
5) Results and discussion: authors did a good job by summarizing the study findings. Despite, I found sometimes the incidence of each complication at 1.5, 3, 6, and 12 years of age are confusing because of the many numbers that is difficult to remember! But the investigators illustrated their findings in a simple but informative figure (1,2 and 3).
Author Response
Author's Reply to the Review Report (Reviewer 2)
Thank you for the opportunity of revising our manuscript and considering it for publication in Children. We would like to thank the reviewers for their constructive comments.
Please find attached our revised manuscript and response point-by-point to the reviewers.
We feel that this article will be appealing to the readership of the journal as it is relevant to CDH and Pediatrics in general.
We confirm that the manuscript has not been published previously.
We look forward to your comments and suggestions and thank you in advance for your consideration.
Thank you very much for the opportunity to review this manuscript that I read with great interest. In this retrospective review of database from JCDHSC, investigators reported long-term outcome of infants diagnosed with CDH either antenatally or during the first 4 weeks after birth. Investigators reported incidence of different complications associated with CDH infants at 1.5, 3, 6, and 12 years of age.
I would like to congratulate the investigators for the well written manuscript
Here are my comments on the manuscript
→Thank you so much for your comments.
1) Title: reflect the study content
2) Abstract: summarizes well the study methodology and results
3) Introduction: concise and indicated aim of the study
4) Methods: No concerns
5) Results and discussion: authors did a good job by summarizing the study findings. Despite, I found sometimes the incidence of each complication at 1.5, 3, 6, and 12 years of age are confusing because of the many numbers that is difficult to remember! But the investigators illustrated their findings in a simple but informative figure (1,2 and 3).
→We apologize for the very difficult readability of the numerous data presented from an epidemiological perspective. We have made it as it is now because we believe that it is necessary for you to see the figures in a simple diagram and to check the numbers when reviewing the details. We hope you will forgive us. Thank you so much.
Sincerely yours,
Masaya Yamoto
Kouji Nagata
